# Analysis of WRKY Gene Family in *Acer fabri* and Their Expression Patterns Under Cold Stress

**DOI:** 10.3390/genes16030344

**Published:** 2025-03-17

**Authors:** Gongwei Chen, Yixiao Zhou, Dandan Zhang, Fengyuan Chen, Xuyang Qin, Hongyu Cai, Heng Gu, Yuanzheng Yue, Lianggui Wang, Guohua Liu

**Affiliations:** 1School of Landscape Architecture, Jiangsu Vocational College of Agriculture and Forestry, No. 19 Wenchang East Road, Jurong 212400, China; chengongwei0118@163.com (G.C.); 13912109566@163.com (H.C.); 2Key Laboratory of Landscape Architecture, College of Landscape Architecture, Nanjing Forestry University, No. 159 Longpan Road, Nanjing 210037, China; 13767185707@163.com (Y.Z.); d2273429918@126.com (D.Z.); chenfengyuan@njfu.edu.cn (F.C.); 13663089792@163.com (X.Q.); guheng@njfu.edu.cn (H.G.); yueyuanzheng@njfu.edu.cn (Y.Y.)

**Keywords:** *Acer fabri*, WRKY gene family, cold stress response

## Abstract

Background/Objectives: The WRKY gene family plays a critical role in plant stress responses; however, its function in *Acer fabri* (*A. fabri*) under cold stress conditions remains poorly understood. This study aims to identify WRKY genes in *A. fabri*, analyze their structural characteristics, and investigate their expression patterns under cold stress, thereby establishing a foundation for further exploration of their roles in cold stress responses. Methods: Using transcriptional data from *A. fabri* subjected to cold stress, we identified 46 WRKY family genes. We employed bioinformatics tools to conduct a comprehensive analysis of the physical and chemical properties of these genes, predict their subcellular localization, and construct a phylogenetic tree. A heatmap was generated to visualize the expression levels of WRKY genes across different treatment conditions. To validate our findings, qRT-PCR was performed on 10 highly expressed WRKY genes to analyze their temporal expression patterns during cold stress exposure. Results: The analysis revealed that WRKY genes in *A. fabri* are predominantly localized to the nucleus, with protein lengths ranging from 55 to 1027 amino acids. Notably, all WRKY genes possessed the conserved WRKYGQK domain. Under cold stress conditions, the WRKY gene expression exhibited a general trend of increasing followed by decreasing, with peak expression observed at 24 h post-treatment. qRT-PCR analysis corroborated this pattern for the selected genes. Conclusions: This study represents the first comprehensive structural and expression analysis of the *A. fabri* WRKY gene family under cold stress conditions. Our findings provide valuable insights into their potential roles in plant cold stress responses, and lay the groundwork for future investigations into the molecular mechanisms underlying WRKY-mediated cold stress tolerance in *A. fabri*.

## 1. Introduction

*Acer fabri (A. fabri)* is a colorful ornamental tree species native to southern China, where its leaves transition from green to red during winter, enhancing landscaping aesthetics [1]. Environmental factors, such as extreme temperatures (both low and high), drought, or excessive moisture, can exert differential impacts on plant physiology. Among these abiotic stresses, cold stress significantly disrupts plant signal transduction pathways, transcriptional regulation, translational processes, and post-translational modifications, ultimately influencing metabolic activities [2]. Cold stress tolerance has long been a central focus in plant stress research, and *A. fabri* is no exception. When exposed to cold stress, plants may experience a range of physiological and biochemical disruptions, including alterations in cellular architecture, the colloidal properties of the protoplasm, water status, osmotic pressure, photosynthetic efficiency, respiratory rates, metabolic pathways, and the enzymatic activity of protective proteins [3,4,5]. These disruptions can impair plant growth and development and, in severe cases, result in plant mortality.

The WRKY transcription factor (TF) family constitutes one of the largest TF families in higher plants, and plays a pivotal role in regulating plant growth, development, and stress responses, particularly cold resistance [6]. Structurally, WRKY TFs are characterized by a highly conserved WRKY domain, with the N-terminal region containing the invariant WRKYGQK sequence and the C-terminal region harboring zinc finger motifs of either C2H2 or C2HC type. Recent studies have underscored the critical role of WRKY TFs in plant cold stress responses. For instance, in *Solanum melongena*, two differentially expressed WRKY genes, *SmWRKY26* and *SmWRKY32*, were shown to respond to cold stress through virus-induced gene silencing (VIGS) technology [7]. In *Forsythia suspensa*, a comprehensive analysis of the entire WRKY family genome revealed that 17 WRKY genes are involved in low-temperature responses, with nine of these genes containing low-temperature-responsive cis-regulatory elements [8]. Furthermore, in *Vaccinium uliginosum*, the overexpression of the *VuWRKY* gene significantly enhanced seedling survival rates, demonstrating its positive role in mitigating cold and salt stress [9].

To date, the WRKY TF family has been extensively investigated in model plants such as Arabidopsis, rice, and wheat. However, research on this family in *A. fabri* remains limited, despite its exceptional ornamental value, adaptability, and notable potential for cold resistance. This characteristic is particularly significant for its introduction and cultivation in colder regions. Given this fact, the present study aims to identify and characterize the WRKY gene family in *A. fabri*, with the objective of providing a robust theoretical foundation for understanding its functional responses and regulatory mechanisms under cold stress conditions. Through this research, we seek to elucidate the pivotal role of the *A. fabri* WRKY TF family in conferring cold resistance, offering novel scientific insights and genetic resources that could enhance the introduction, cultivation, and landscaping applications of this species. This study not only addresses the existing gap in knowledge regarding the WRKY TF family in *A. fabri,* but also aims to provide new perspectives for its future development and utilization.

## 2. Materials and Methods

### 2.1. Plant Materials

The experimental samples consisted of *A. fabri* trees, with all plant materials adhering to relevant institutional, national, and international standards and regulations. Specifically, three five-year-old *A. fabri* plants were utilized, cultivated at the Jiangsu Agricultural Expo Park located at latitude 32°01′ and longitude 119°24′. During the experiment, potted *A. fabri* plants were subjected to cold treatment in a growth chamber set at 4 °C. Leaf samples were collected at 0, 3, 6, 12, 24, and 48 h post-treatment. Immediately following collection, the leaves were placed into sterile tubes and frozen in liquid nitrogen for 10 min before being transferred to a −80 °C ultra-low temperature freezer for long-term storage. Biological replicates were performed three times for each temperature treatment phase.

### 2.2. qRT-PCR Gene Expression Validation

Total RNA was isolated from the leaves of *A. fabri* using the RNA Purification Kit (Invitrogen, Carlsbad, CA, USA). The extracted RNA was reverse-transcribed into cDNA using the SuperMix Reverse Transcription Kit (Transgen, Beijing, China) according to the manufacturer’s instructions [10]. The synthesized cDNA was then diluted 20-fold for subsequent gene expression analysis. Real-time quantitative PCR (qRT-PCR) was performed using the SYBR Premix Ex Taq kit (Takara Biotechnology, Nanjing, China) to validate the expressions of selected key genes and TFs. The qRT-PCR conditions were as follows: an initial pre-denaturation step at 95 °C for 3 min, followed by 40 cycles of denaturation at 95 °C for 5 s and annealing/extension at 60 °C for 30 s. Primer design was accomplished using Primer 5 software, and the sequences are listed in Appendix A. The TUB gene of *Acer buergerianum* was chosen as the reference gene for normalization [11]. Relative gene expression levels were calculated using the 2^−∆∆Ct^ comparative Ct method [12]. Each sample was analyzed with 3 technical replicates and 3 biological replicates to ensure statistical reliability.

### 2.3. De Novo Assembly of RNA-Seq Reads and Quantitative Analysis of Gene Expression

To ensure accurate gene expression analysis, initial steps involved removing adapter sequences from raw sequencing reads. Quality control was performed using fastp v0.18.0, where reads with quality scores below 130 (comprising >40% of the total bases) and reads containing >10% unknown bases were removed. High-quality reads from all samples were subsequently pooled together. The alignment of these clean reads to the reference genome was conducted using HISAT v2.2.4. Following alignment [12,13], reads were assembled according to the guidelines of StringTie v1.3.1 [14,15]. Gene expression levels were quantified using a fragment per kilobase of transcript per million mapped reads (FPKM)-based exon model, which accounts for gene length and sequencing depth variations. Transcripts with fold change >2 or <−2 and a Q value ≤ 0.05 are considered differentially expressed genes [16].

### 2.4. RNA Extraction, cDNA Library Construction, and Sequencing

Total RNA was extracted from nine samples using the RNA Purification Kit 110 (Invitrogen, Carlsbad, CA, USA). RNA integrity was assessed via RNase-free agarose gel electrophoresis, and RNA concentration was determined using a NanoDrop 2000 spectrophotometer (Thermo Fisher Scientific, Waltham, MA, USA) [17]. Enriched mRNA was fragmented using fragment buffer, and subsequent reverse transcription generated cDNA using random primers. The second strand of cDNA was synthesized employing DNA polymerase I, RNase H, dNTPs, and buffer. The resultant cDNA fragments were purified using the QIAquick RT-PCR Extraction Kit (Qiagen, Venlo, The Netherlands), followed by end repair, poly(A) addition, and the ligation of Illumina sequencing adapters [18]. A cDNA library targeting the color change of *A. fabri* leaves was constructed using three RNA samples and sequenced on the Illumina NovaSeq 6000 platform by Gene Denovo Biotech Co. (Guangzhou, China). The transcriptome data have been deposited in the NCBI Sequence Read Archive (SRA) under the accession number PRJNA1150814.

### 2.5. Measurement of Physiological Indicators

This study analyzed the physiological responses of *A. fabri* leaves exposed to low temperature stress at 4 °C. Levels of proline (Pro; Jiancheng, Nanjing, China, Cat# A003-1-1), peroxidase (POD; Cat# A084-3-1), and superoxide dismutase (SOD; Cat# A001-3-2) were measured following manufacturers’ protocols. Soluble sugar (SS; Beijing Bokebio, Beijing, China, Cat# BC0035) and soluble protein (SP; Cat# BC0021) contents were analyzed with colorimetric assays [19]. All procedures strictly followed the manufacturers’ protocols.

### 2.6. Identification of Members of the A. fabri WRKY Gene Family

WRKY TF sequences were identified from the transcriptome sequencing data of *A. fabri* [1]. Full-length amino acid sequences of WRKY TFs were obtained through the NCBI ORF Finder tool. A Blast alignment was performed between these sequences and known members of the Arabidopsis WRKY gene family, with results filtered for an E-value threshold of <1 × 10^−5^. The intersection of alignment results was taken as the preliminary set of *A. fabri* WRKY TF family members. Domain integrity and accuracy were confirmed by submitting these sequences to NCBI Batch CD-Search and SMART for domain analysis [4]. After validation, these genes were designated as members of the *A. fabri* WRKY TF family.

### 2.7. Bioinformatics Analysis of the WRKY Gene Family in A. fabri

The physicochemical properties of WRKY family proteins in *A. fabri* were analyzed using the ProtParam online tool [20]. The subcellular localization of these proteins was predicted using Cell-PLoc 2.0 [21]. Conserved motifs within the WRKY proteins were identified using MEME, and motif visualization was performed using TBtools software No.2.007 [22]. To construct a phylogenetic tree for the WRKY TF family in *A. fabri*, MEGA 7.0 was utilized with the neighbor-joining (NJ) method, incorporating 1000 bootstrap replicates to ensure robust results [23,24].

### 2.8. Expression Analysis of A. fabri WRKY Genes

Expression data for WRKY family genes were extracted from the transcriptome data of *A. fabri*, and their fragments per kilobase million (FPKM) values were selected for further analysis [1]. Gene expression levels were visualized using a heatmap generated with TBtools software.

## 3. Results

### 3.1. General Characteristics of the A. fabri WRKY Gene Family

From the cold stress transcriptomic data of *A. fabri*, 46 candidate WRKY gene sequences were identified. These WRKY gene family members underwent physicochemical property analysis and subcellular localization prediction (Appendix A). The results reveal that the lengths of the 46 *A. fabri* WRKY proteins range from 55 to 1027 amino acids, with molecular weights varying from 6496.31 to 111,962.45 Da. The longest protein, Unigene0016335, and the shortest protein, Unigene0024195, represent the extremes of this range. The theoretical isoelectric points (pI) of these proteins range from 4.73 (Unigene0080799) to 10.33 (Unigene0045308), and the instability indices range from 32.07 to 77.09. All proteins exhibit negative hydrophilicity values (−1.433 to −0.308), indicating their hydrophilic nature. Subcellular localization prediction confirmed that all WRKY proteins were localized in the nucleus.

### 3.2. Analysis of the Conserved Domains of the WRKY Gene Family in A. fabri

Conserved domains of the *A. fabri* WRKY gene family were analyzed using the MEME Suite (Figure 1). Ten conserved motifs (Motif1–Motif10) were identified, with Motif1 and Motif3 being characteristic of WRKY TFs, and Motif8 corresponding to the zinc finger domain. Most family members contain Motif1 and Motif2; however, three exceptions—Unigene0036507, Unigene0064835, and Unigene0074140—lack these motifs. In contrast, Unigene0026322 and Unigene0050758 retain both motifs, and the majority of family members possess three to six motifs.

### 3.3. Evolutionary Analysis of the WRKY Gene Family

A phylogenetic tree was constructed using Mega7 software to analyze the evolutionary relationships among 46 members of the *A. fabri* WRKY gene family and the WRKY gene family domains of *Arabidopsis thaliana* (Figure 2). The analysis revealed distinct groupings within the WRKY family, as follows: 12 members belonged to Class I, which formed a cohesive cluster; Class II-C contained 13 members; and Class II-B included 5 members, with 2 forming a separate subgroup. Additionally, Class III comprised six members that grouped together. Within Class II-E, the 10 members were divided into two distinct clusters—one containing 3 members and the other containing 7 members.

### 3.4. Expression Analysis of the WRKY Gene Family Members Under Cold Stress

An analysis of the expression patterns of the *A. fabri* WRKY genes (Figure 3) revealed distinct responses to cold stress. Specifically, *Unigene0026320* and *Unigene0036753* exhibited strong activation responses following a 48 h cold treatment, with their expression levels continuing to rise throughout the duration. Compared to other WRKY family members, these two genes showed significantly higher expression levels. Additionally, *Unigene0026322* and *Unigene0036509* were activated after 6 h of cold treatment; however, their expression declined as the treatment duration extended to 48 h. The expression of *Unigene0047036* reached its peak at 24 h but began to decline by the 48 h mark.

### 3.5. The Correlation Analysis Between AfWEKYs Gene Expression and Physiological Indicators

The physiological traits of *A. fabri* were analyzed over a 48 h period (from 0 h to 48 h). A correlation heatmap illustrating the relationship between the *A. fabri* WRKY gene family and physiological traits is presented in Figure 4. The results indicate that most WRKY genes in *A. fabri* exhibited significant positive correlations with proline (Pro), soluble sugars (SS), and soluble proteins (SP). Specifically, the genes *Unigene0026320*, *Unigene0026321*, *Unigene0026322*, *Unigene0036509*, *Unigene0036753*, *Unigene0064835*, *Unigene0075318*, and *Unigene0078068* demonstrated significant positive correlations with superoxide dismutase (SOD) and peroxidase (POD). Additionally, *Unigene0047036* and *Unigene0048493* were positively correlated with proline (Pro), soluble sugars (SS), and soluble proteins (SP).

### 3.6. Real-Time Fluorescence Quantitative PCR (qRT-PCR) Verification of the WRKY Gene Family Members in A. fabri

To validate the transcriptomic data, ten WRKY TFs with relatively high expression levels under cold stress were selected for real-time fluorescence quantitative PCR (qRT-PCR) analysis (Figure 5). The results demonstrate that the expression profiles of most genes followed a pattern of an initial increase followed by a decrease. Specifically, the expression levels of *Unigene0026320*, *Unigene0036753*, *Unigene0047036*, *Unigene0075318*, and *Unigene0078068* reached their peaks at 24 h, while *Unigene0026321*, *Unigene0026322*, *Unigene0036509*, and *Unigene0064835* achieved maximum expression at 12 h. Notably, the expression level of *Unigene0048493* remained stable from 3 h to 24 h.

## 4. Discussion

### 4.1. Functional Divergence of the WRKY Gene Family in A. fabri and Other Woody Plants

The WRKY gene family of *A. fabri* exhibits a smaller size compared to that of poplar (101 members) and eucalyptus (87 members) [25,26], which may reflect its reduced stress complexity within its ecological niche (temperate deciduous forest). For instance, poplar WRKY genes demonstrate high differentiation in wood development and insect resistance, while *A. fabri* WRKY genes are primarily associated with abiotic stress responses (e.g., cold and drought), underscoring differences in functional adaptability among woody plants [27,28,29]. A comparative analysis with *Acer pseudosieboldianum* revealed that the unique *Unigene0026509* (homologous to *AtWRKY7*) in *A. fabri* shows an 8-fold increase in expression under cold stress [30], whereas the homologous gene in *Acer truncatum* is only 2-fold upregulated. This observation suggests that *A. fabri* may possess an enhanced capacity to perceive low-temperature signals [31]. Additionally, the molecular mechanism underlying WRKY gene activation during cold stress demonstrates rapid response kinetics and unique adaptive features.

### 4.2. Functional Verification and Regulation Network of A. fabri

Functional verification via qRT-PCR has demonstrated that *AfWRKY33* (*Unigene0026320*) exhibits robust multi-stress cross-regulation properties. Specifically, its expression increases 15-fold, 7-fold, and 3-fold under cold, drought, and salt stress, respectively, while *MbWRKY33* responds exclusively to drought [32]. Experimental evidence indicates that WRKY33 directly binds to the W-box element in the promoter of the lipid metabolism gene GPAT6, thereby enhancing cold tolerance by stabilizing membrane lipids [33]. Furthermore, studies on tomatoes have shown that *SlWRKY33* directly targets and induces the expressions of genes such as CDPK11, MYBS3, and BAG6, thereby improving cold tolerance [34]. In plum blossoms, the overexpression of *PmWRKY57* enhances cold resistance in *A*. *thaliana* [35]. Similarly, WRKY TFs in *F. suspensa* are upregulated under cold stress [36].

### 4.3. Evolutionary Drivers of Functional Diversification in the A. fabri WRKY Gene Family

This study provides novel insights into the unique adaptive characteristics of the *A. fabri* WRKY family in woody plants. Unlike species with larger genomes, such as poplar and eucalyptus, *A. fabri* achieves functional intensification through limited gene duplication (e.g., *AfWRKY33*/*AfWRKY31* paralogs), which may reflect its shorter life cycle and adaptation to temperate environments. The discovery of WRKY-bZIP chimeric proteins offers new evidence for TF functional diversity in woody plants [37], potentially enhancing stress signal integration through multi-domain synergy. Notably, the rapid activation of *A. fabri* WRKY genes (peaking within 6 h) surpasses that of most Rosaceae (e.g., apples) and Solanaceae plants [7,32], likely due to natural selection pressures derived from its high-altitude habitat. The WRKY family plays a pivotal role in plant growth, development, and abiotic stress responses [6,38,39]. In *A. fabri*, we identified 46 WRKY genes, comparable to other plants such as Arabidopsis (74 WRKY genes) [40], rice (51 WRKY genes) [41], and potatoes (82 WRKY genes) [42,43]. Our findings suggest that the *A. fabri* WRKY family is critical for cold tolerance, and further support the finding of functional characteristics of WRKY TFs in Acer species.

## 5. Conclusions

The WRKY TF family in *A. fabri* demonstrates unique functional and evolutionary adaptations shaped by its temperate deciduous habitat. Our analysis of 20 WRKY TFs in *A. fabri* suggests their potential role in the cold tolerance regulatory network of this species. Future studies will explore the functional divergence and regulatory mechanisms of WRKY gene families in *A. fabri*, further elucidating their contribution to environmental adaptation and molecular breeding in temperate woody plants.

## Figures and Tables

**Figure 1 genes-16-00344-f001:**
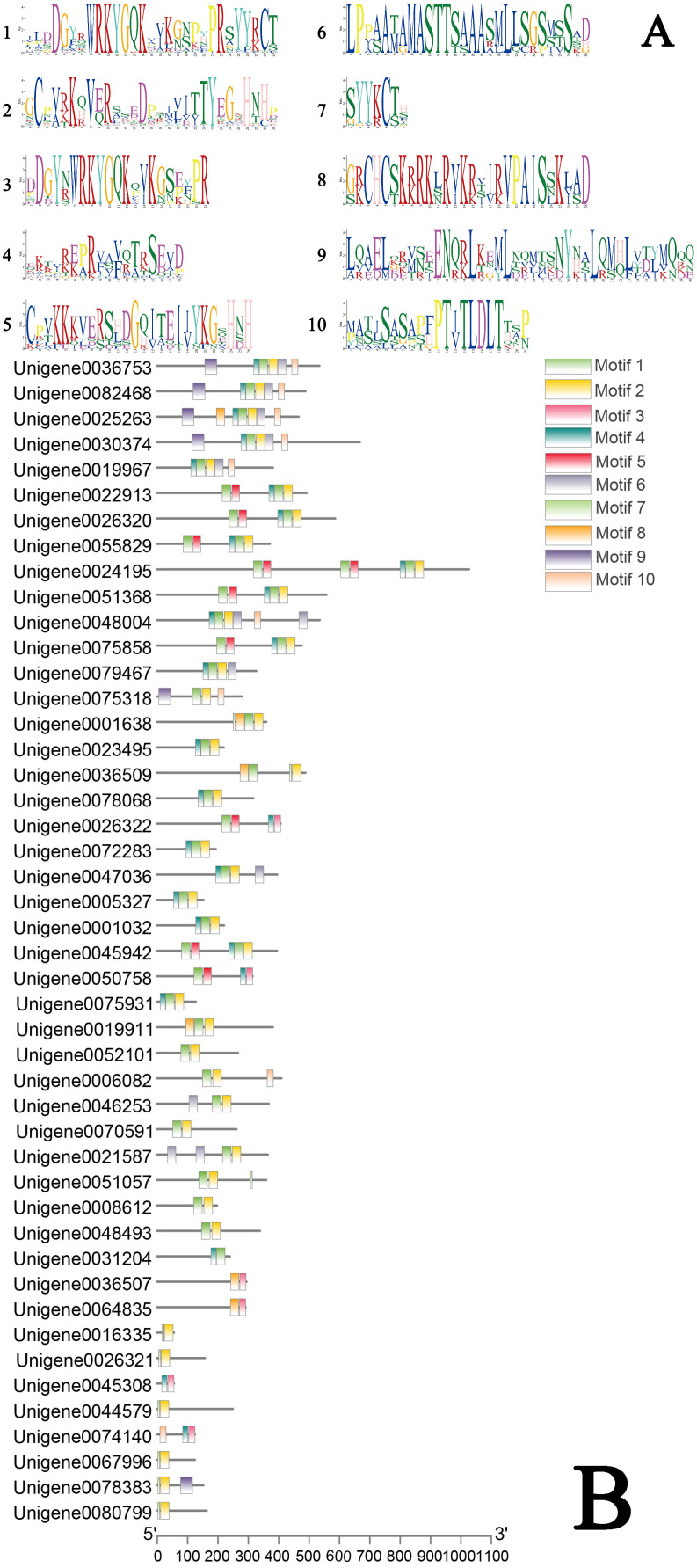
Analysis of the conserved structure of the WRKY gene family in the *A. fabri* ((**A**) *A. fabri’*s motif; (**B**) The motif location of *A. fabri*).

**Figure 2 genes-16-00344-f002:**
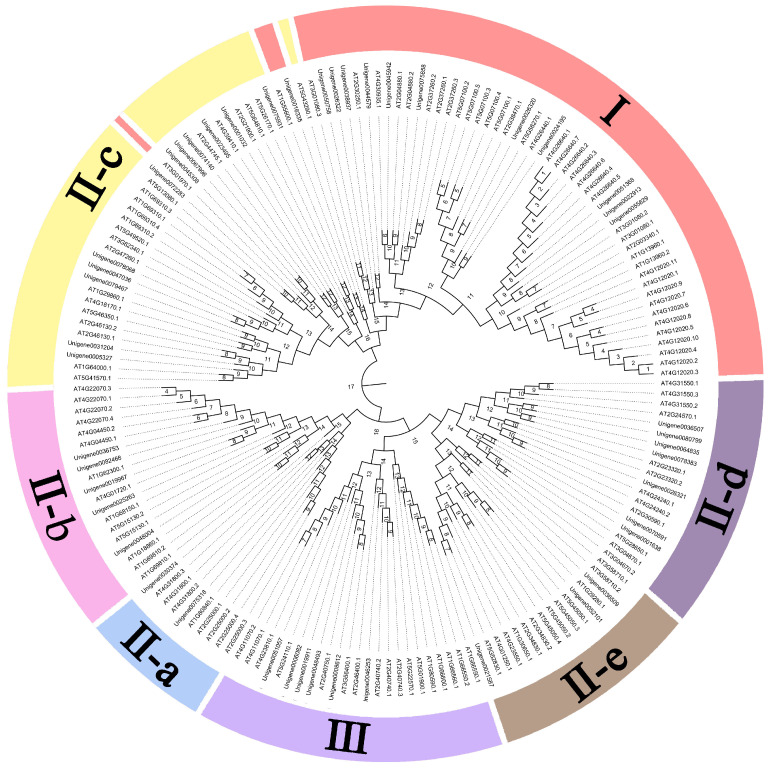
Phylogenetic tree of *A. fabri* WRKY proteins constructed using the Neighbor-Joining method in MEGA 7.0 with 1000 bootstrap replicates (values ≥ 50% shown). The JTT substitution model and pairwise deletion of gaps were applied. Scale bar: 0.2 substitutions per site.

**Figure 3 genes-16-00344-f003:**
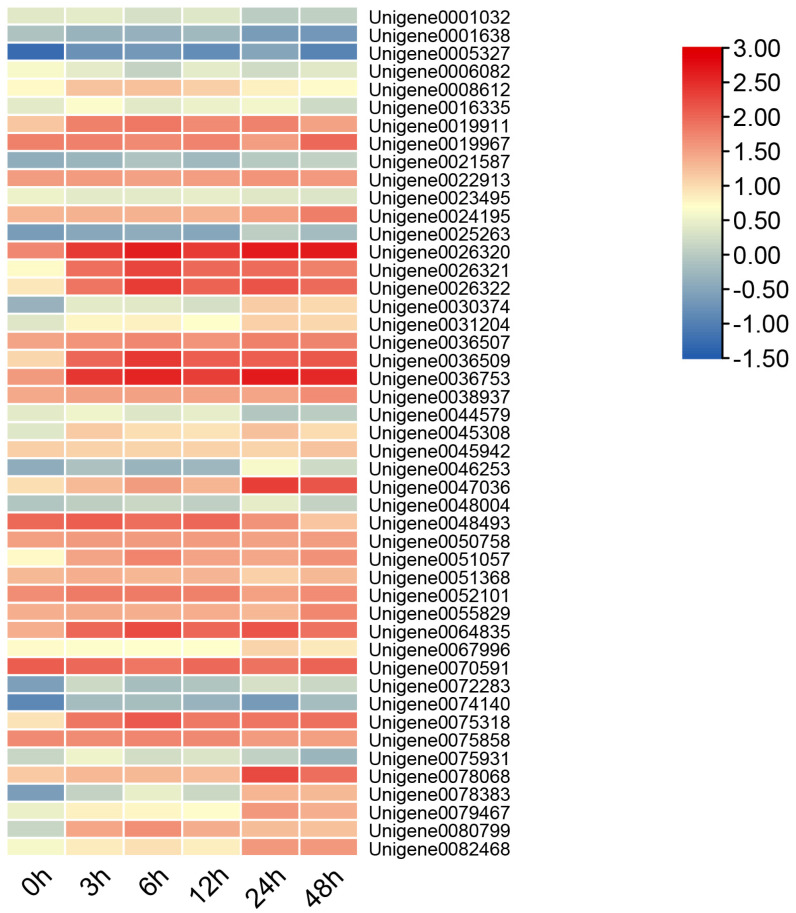
Heatmap of gene expression of the *A. fabri* WRKY family under 48 h of cold treatment.

**Figure 4 genes-16-00344-f004:**
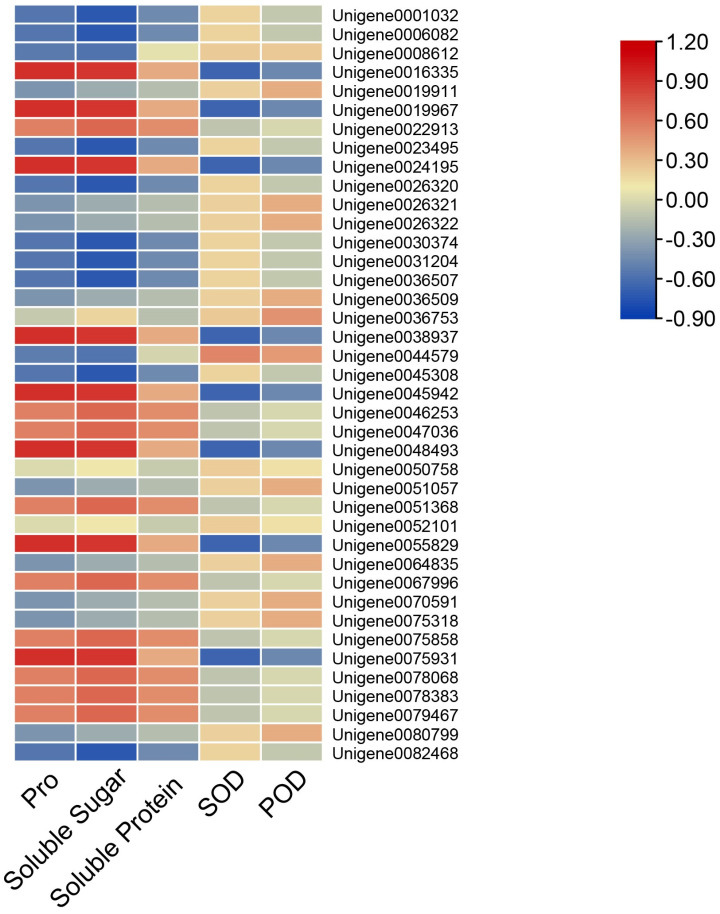
Correlation analysis between 5 physiological indicators and the expression level of *AfWRKYs*.

**Figure 5 genes-16-00344-f005:**
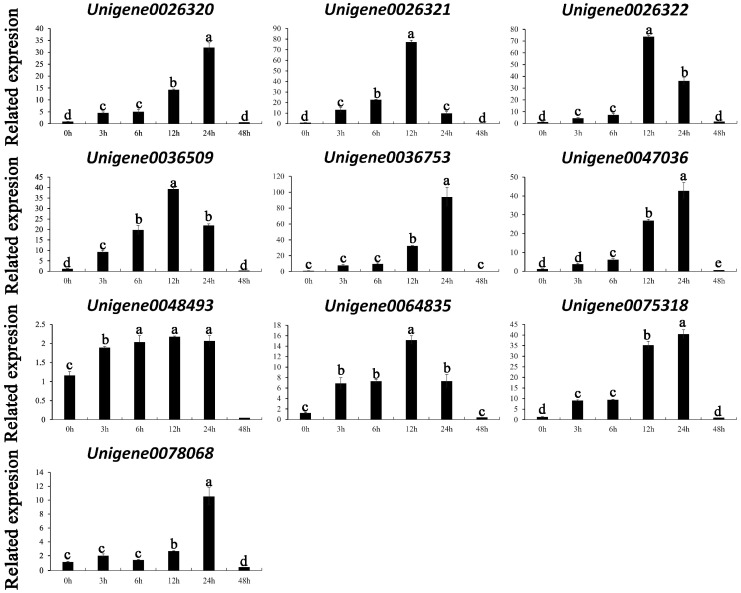
qRT-PCR results of the *A. fabri* WRKY family members.

## Data Availability

Sequence data that support the findings of this study are available in the NCBI Sequence Read Archive under accession code PRJNA1150814 (https://www.ncbi.nlm.nih.gov/sra/ accessed on 15 January 2024).

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
