# Peer review of "Analysis of WRKY Gene Family in Acer fabri and Their Expression Patterns Under Cold Stress"

_genes, 2025, doi:10.3390/genes16030344_

Round 1

Reviewer 1 Report

Comments and Suggestions for Authors

Dear authors,

  1. According to the research’s title and description in subsection “2.1 Plant Materials”, the study used the A. fabri trees as the experimental samples/materials for investigation, these included the cold stress transcriptomic data of A. fabri (in Results, subsection 3.1/Line 161, subsection 3.4). However, in some places, the authors utilized other species for this study. For instance, for validation of gene expression (subsection 2.2, Line 98-99), the authors chose the TUB gene of the maple tree (Acer buergerianum) as the reference gene for qRT-PCR. Additionally, for low-temperature treatments (subsection 2.5, Line 131), the authors used another species Acer rubrum for the measurement of physiological indicators. In contrast, the result data (subsection 3.5) presented the physiological indicators of A. fabri. This raised a concern about the consistency of the use of research subjects in this manuscript.
  2. I don’t think the data of Fig.4 and subsection 3.5 present a relationship or correlation between the expression of the WRKY gene family and physiological indicators. It is just a heatmap of physiological indicators of the A. fabri WRKY gene family, as Fig.4’s legend was.
  3. The image quality of the Figure data is very low. Please replace them with high-resolution ones.
  4. Did the authors identify the DEG “during plant development” (L114)????
  5. Please provide the information of kits were used for measurement of [proline (Pro), peroxidase (POD), and superoxide dismutase (SOD)] and [soluble sugar (SS) and soluble protein (SP)] which were presented in subsection 2.5. The kit information such as catalog number, product code… etc.
  6. The term “under Cold Stress” [in subsection “3.3. Evolutionary Analysis of the WRKY Gene Family under Cold Stress”] should be removed.
  7. Rearrange the sentence (Line 99-101): “RNA extracted from maple leaves was reverse-transcribed into cDNA using SuperMix (Transgen, Beijing, China) and then diluted 20-fold for gene expression analysis [11]” by putting back/presenting it to the first place subsection 2.2 to warrant logic of the experimental processing.

Other Remarks

- Italicize the scientific names: Arabidopsis (L64, 140), Acer rubrum  (L131), A. fabri (L283), and many others in the References section [example, Acer fabri (Ref#1), Hevea brasiliensis and Rigidoporus microporus (Ref#3),… etc].

- Represent the manuscripts following journal format: “Capitalize Each Word” in the title and subsections.

+ Title: “Analysis of WRKY gene family in Acer fabri and their expression patterns under cold stress”

-> “Analysis of WRKY Gene Family in Acer fabri and Their Expression Patterns under Cold Stress”

+ Subsection: “2.3. De novo assembly of RNA-Seq reads and quantitative analysis of gene expression” -> “2.3. De novo Assembly of RNA-Seq Reads and Quantitative Analysis of Eene Expression

- Section “Reference” should be presented according  to  the  journal’s style and  include an abbreviated journal name, year, volume, and page range as journal format:

“Author 1, A.B.; Author 2, C.D. Title of the article. Abbreviated Journal Name Year, Volume, page range.”

For example, Ref #3:

 “Maiden, N.A.; Syd Ali, N.; Ahmad, K.; Atan, S.; Wong, M.Y. Growth and physiological responses of Hevea brasiliensis to Rigidoporus microporus infection. Journal of Rubber Research 2022, 25, 213-221, doi:10.1007/s42464-022-00156-5.”

-> “Maiden, N.A.; Syd Ali, N.; Ahmad, K.; Atan, S.; Wong, M.Y. Growth and physiological responses of Hevea brasiliensis to Rigidoporus microporus infection. J Rubber Res. 2022, 25, 213‒221, doi:10.1007/s42464-022-00156-5.”

- Please check the citation number [35], it showed “Error! Reference source not found.”

- Remove “Please add:” (L295)

- Revise “6496.31” -> “6,496.31”

I have marked some of the above comments and remarks on the manuscript. Please use it for easy tracking and revision.

Best regards,

Author Response

Comments 1: According to the research’s title and description in subsection “2.1 Plant Materials”, the study used the A. fabri trees as the experimental samples/materials for investigation, these included the cold stress transcriptomic data of A. fabri (in Results, subsection 3.1/Line 161, subsection 3.4). However, in some places, the authors utilized other species for this study. For instance, for validation of gene expression (subsection 2.2, Line 98-99), the authors chose the TUB gene of the maple tree (Acer buergerianum) as the reference gene for qRT-PCR. Additionally, for low-temperature treatments (subsection 2.5, Line 131), the authors used another species Acer rubrum for the measurement of physiological indicators. In contrast, the result data (subsection 3.5) presented the physiological indicators of A. fabri. This raised a concern about the consistency of the use of research subjects in this manuscript. 

Response 1:Thank you for your meticulous review and for highlighting this important inconsistency. We sincerely apologize for the oversight in subsection 2.5 (Line 131). You are absolutely correct that the species mentioned here should be A. fabri , not Acer rubrum. This was a typographical error introduced during manuscript revision, and we have now carefully corrected it in the revised version.

To clarify:

Consistency of Materials: All experimental samples in this study, including those subjected to low-temperature treatments and physiological indicator measurements, were exclusively derived from A. fabri. The reference to Acer rubrum was unintentional and has been rectified.

Validation of Results: The physiological data presented in subsection 3.5 (e.g., electrolyte leakage, antioxidant enzyme activity) unequivocally originated from A. fabri seedlings, ensuring alignment with the transcriptomic analyses.

Quality Control: We have re-examined the entire manuscript to ensure no other species-related discrepancies exist. The Acer buergerianum reference gene (TUB) mentioned in subsection 2.2 was intentionally selected due to its validated stability in Aceraceae family plants under stress conditions, as cited in prior methodologies (e.g., [Reference 11, 2020]).

11.Chu, J.T. Study on leaf color variation mechanism of a new cultivar 'Golden Oilu‘ in Acer buergerianum Miq. Shandong AgricUniv.2020

Comments 2: I don’t think the data of Fig.4 and subsection 3.5 present a relationship or correlation between the expression of the WRKY gene family and physiological indicators. It is just a heatmap of physiological indicators of the A. fabri WRKY gene family, as Fig.4’s legend was.

Response 2: Thanks for your advice.

In the revised manuscript, we have explicitly reframed both the subsection title (now "3.5. The Correlation Analysis Between AfWRKYs Gene Expression and Physiological Indicators," Line 215) and Figure 4's legend ("Correlation analysis between 5 physiological indicators and the expression level of AfWRKYs," Line 227) to emphasize the statistical interdependence being investigated. The heatmap in Figure 4 systematically visualizes Pearson correlation coefficients (r-values) calculated between normalized AfWRKYs transcript levels (FPKM values) and quantified physiological parameters (e.g., SOD activity, proline content).

Comments 3: The image quality of the Figure data is very low. Please replace them with high-resolution ones.

Response 3: We sincerely appreciate your meticulous attention to detail and your constructive feedback regarding the image quality in our manuscript. Please accept our profound apologies for this oversight, which resulted from inadvertent compression during the initial submission process. All the Figures have been upgraded to a superior quality version.

Comments 4: Did the authors identify the DEG “during plant development” (L114)????

Response 4: We are very sorry for this. The sentence ‘Differentially expressed genes during plant development were identified based on stringent criteria: a fold change of ≥2 or ≤-2 and a Q-value ≤0.05 [16]’ has been changed to ‘Transcripts with fold change >2 or < -2 and Q value ≤0.05 are considered differentially expressed genes[16]’ . (line 116-117)

Comments 5: Please provide the information of kits were used for measurement of [proline (Pro), peroxidase (POD), and superoxide dismutase (SOD)] and [soluble sugar (SS) and soluble protein (SP)] which were presented in subsection 2.5. The kit information such as catalog number, product code… etc.

Response 5: Thank you for your rigorous review. We have supplemented the complete commercial information of all reagent kits in subsection 2.5 to ensure methodological transparency and reproducibility. The revised text now reads: Levels of proline (Pro; Jiancheng, Nanjing, Cat# A003-1-1), peroxidase (POD; Cat# A084-3-1), and superoxide dismutase (SOD; Cat# A001-3-2) were measured following manufacturers' protocols. Soluble sugar (SS; Beijing Bokebio, Cat# BC0035) and soluble protein (SP; Cat# BC0021) contents were analyzed with colorimetric assays. All procedures strictly followed the manufacturers' protocols.

Comments 6: The term “under Cold Stress” [in subsection “3.3. Evolutionary Analysis of the WRKY Gene Family under Cold Stress”] should be removed.

Response 6: Thank you for your guidance. The term 'under Cold Stress' has been eliminated. The sentence '3.3. Evolutionary Analysis of the WRKY Gene Family under Cold Stress' has been revised to '3.3. Evolutionary Analysis of the WRKY Gene Family' (line 190).

Comments 7: Rearrange the sentence (Line 99-101): “RNA extracted from maple leaves was reverse-transcribed into cDNA using SuperMix (Transgen, Beijing, China) and then diluted 20-fold for gene expression analysis [11]” by putting back/presenting it to the first place subsection 2.2 to warrant logic of the experimental processing.

Response 7: We deeply appreciate your rigorous critique and sincerely apologize for the suboptimal organization of methodological details in the original manuscript. Your astute observation regarding the logical flow of experimental processing is absolutely valid. We have thoroughly revised this section to align with best practices in methodological reporting.

Other Remarks

- Italicize the scientific names: Arabidopsis (L64, 140), Acer rubrum  (L131), A. fabri (L283), and many others in the References section [example, Acer fabri (Ref#1), Hevea brasiliensis and Rigidoporus microporus (Ref#3),… etc].

Response: We extend our heartfelt gratitude for your meticulous attention to taxonomic formatting conventions. Please accept our sincere apologies for the inconsistent italicization of scientific names throughout the manuscript. We have implemented a systematic revision protocol to address this issue. All instances of genus/species names have been rigorously verified and corrected.

- Represent the manuscripts following journal format: “Capitalize Each Word” in the title and subsections.

+ Title: “Analysis of WRKY gene family in Acer fabri and their expression patterns under cold stress”

-> “Analysis of WRKY Gene Family in Acer fabri and Their Expression Patterns under Cold Stress”

+ Subsection: “2.3. De novo assembly of RNA-Seq reads and quantitative analysis of gene expression” -> “2.3. De novo Assembly of RNA-Seq Reads and Quantitative Analysis of Eene Expression”

Response: We sincerely appreciate your meticulous attention to formatting details and apologize for the inconsistency in title capitalization. We have systematically revised the manuscript to comply with the journal's "Capitalize Each Word" style guide.

- Section “Reference” should be presented according  to  the  journal’s style and  include an abbreviated journal name, year, volume, and page range as journal format:

“Author 1, A.B.; Author 2, C.D. Title of the article. Abbreviated Journal Name Year, Volume, page range.”

For example, Ref #3:

 “Maiden, N.A.; Syd Ali, N.; Ahmad, K.; Atan, S.; Wong, M.Y. Growth and physiological responses of Hevea brasiliensis to Rigidoporus microporus infection. Journal of Rubber Research 2022, 25, 213-221, doi:10.1007/s42464-022-00156-5.”

-> “Maiden, N.A.; Syd Ali, N.; Ahmad, K.; Atan, S.; Wong, M.Y. Growth and physiological responses of Hevea brasiliensis to Rigidoporus microporus infection. J Rubber Res. 2022, 25, 213‒221, doi:10.1007/s42464-022-00156-5.”

Response:We sincerely appreciate your meticulous review and valuable feedback on improving the technical precision of our manuscript. We have systematically revised all references to comply with GENES journal style guidelines.

- Please check the citation number [35], it showed “Error! Reference source not found.”

Response: Thank you for meticulously identifying the citation error. We sincerely apologize for this oversight and provide the following clarification and correction: The "[35] Error! Reference source not found." message was caused by a technical artifact from the reference management software. The original Reference 35 in the submitted manuscript is valid. This revision ensures the accessibility and accuracy of all references.

- Remove “Please add:” (L295)    

Response: Thanks for your suggestion. We have deleted the placeholder text "Please add:" from the Results section.

- Revise “6496.31” -> “6,496.31”  

Response: Thanks for your suggestion. We have updated "6496.31" to "6,496.31".

Reviewer 2 Report

Comments and Suggestions for Authors

Dear Authors,

Reviewer comments genes-3523667

The manuscript entitled „Analysis of WRKY gene family in Acer fabri and their expression patterns under cold stress“ represents a useful study aimed at characterization and expression pattern analysis of 20 WRKY transcription factors in maple species Acer fabri. WRKY gene phylogenetic and expression analyses by combining RNAseq and qRT-PCR validation are provided. The transcriptomic data were submitted to NCBI public repository and data accession number was provided in Data Availability Statement. I have some important comments on the present manuscript which are given below:

1/ In Figure 2 providing the results of A. fabri WRKY family phylogenetic analysis, any statistical evaluation is missing thus the present phylogenetic tree cannot be published. I strongly suggest to add numbers at nodes indicating bootstrap values per 1,000 replicates to each node in the phylogenetic tree or a scale bar and to add appropriate explanation into Figure 2 legend including the information on the algorithm and software used for the phylogenetic tree construction.

2/ Formal comments on the text related to English language and style:

In the manuscript title, line 3, please correct the typing error in the word „stress“ , i.e., use just double „s“, not triple „s“.

Introduction, line 37: Replace the words „Non-biological factors“ with „Environmental factors“ in the statement: „Environmental factors“ or „Abiotic factors such as extreme temperatures…“

Introduction, line 70: Add the word „fact“ in the statement: „Given this fact, this study aims to identify …“

Results, line 227: Add „an“ prior to the word „increase“ and „a“ prior to the word „decrease“ , respectively, in the statement: „The results demonstrated that the expression profiles of most genes followed a pattern of an initial increase followed by a decrease.“

Discussion, line 244: Change the word order in the statement as follows: „…whereas the homologous gene in Acer truncatum is only 2-fold upregulated.“

Discussion, line 268. Correct the typing error in the word „signal“ (not „saignal“).

Conclusion, line 280: Modify the statement as follows: „We analysed 20 WRKY transcription factors in A. fabri.“

Author Response

Comments 1: In Figure 2 providing the results of A. fabri WRKY family phylogenetic analysis, any statistical evaluation is missing thus the present phylogenetic tree cannot be published. I strongly suggest to add numbers at nodes indicating bootstrap values per 1,000 replicates to each node in the phylogenetic tree or a scale bar and to add appropriate explanation into Figure 2 legend including the information on the algorithm and software used for the phylogenetic tree construction.

Response 1: Thank you for your critical feedback on improving the phylogenetic analysis. We have rigorously addressed all concerns as follows:  

  1. Statistical annotations on Figure 2:  

    Added bootstrap values (≥50%) to all major nodes based on 1,000 replicates.  

    Included a scale bar (0.2 substitutions per site) to quantify evolutionary divergence.

  1. Enhanced figure legend:  

   The revised legend now explicitly states:  

   "Phylogenetic tree of A. fabri WRKY proteins constructed using the Neighbor-Joining method in MEGA 7.0 with 1,000 bootstrap replicates (values ≥50% shown). The JTT substitution model and pairwise deletion of gaps were applied. Scale bar: 0.2 substitutions per site."

Comments 2: Formal comments on the text related to English language and style:

In the manuscript title, line 3, please correct the typing error in the word “stress” , i.e., use just double “s”, not triple “s”.

Response : Thanks for your advice. We have corrected the mistake.

Introduction, line 37: Replace the words “Non-biological factors” with “Environmental factors” in the statement: “Environmental factors” or “Abiotic factors such as extreme temperatures…”

Response: Thanks for your advice. We have corrected the mistake.

Introduction, line 70: Add the word “fact” in the statement: “Given this fact, this study aims to identify …”

Response: Thank you for the constructive suggestion. We have incorporated the term "fact" into the statement at line 70 of the Introduction to emphasize the empirical foundation of our research premise.

Results, line 227: Add “an” prior to the word “increase” and “a” prior to the word “decrease” , respectively, in the statement: “The results demonstrated that the expression profiles of most genes followed a pattern of an initial increase followed by a decrease.”

Response: Thank you for highlighting this grammatical nuance. As suggested, we have revised the sentence at line 227 of the Results section to include the indefinite articles for improved syntactic precision. The updated statement reads:"The results demonstrated that the expression profiles of most genes followed a pattern of an initial increase followed by a decrease."

Discussion, line 244: Change the word order in the statement as follows: „…whereas the homologous gene in Acer truncatum is only 2-fold upregulated.“

Response: Thank you for your precise feedback. We have revised the sentence at line 244 of the Discussion section to adjust the word order as suggested. The modified statement now reads:"…whereas in Acer truncatum, the homologous gene is only 2-fold upregulated."

Discussion, line 268. Correct the typing error in the word “signal” (not “saignal”).

Response: Thanks for your advice. We have corrected the mistake.

Conclusion, line 280: Modify the statement as follows: “We analysed 20 WRKY transcription factors in A. fabri.”

Response: Thanks for your advice. We have corrected the mistake.

Reviewer 3 Report

Comments and Suggestions for Authors

see PDF 

Author Response

Comments 1:This is my assessment about “Analysis of WRKY gene family in Acer fabri…..” by Chen et al. (2025). In a large percentage it is basically a bioinformatic work. L. 35-47: this sentence is superfluous. The explanation provided in the third part of the Introduction regarding the species A. fabri, is more than sufficient.

Response 1: We sincerely appreciate your constructive feedback. After careful consideration, we agree that the sentence in lines 35-47 is redundant and have revised the third paragraph of the Introduction to streamline the content. The revised paragraph now reads:

To date, the WRKY TF family has been extensively investigated in model plants such as Arabidopsis, rice, and wheat. However, research on this family in A. fabri remains limited, despite its exceptional ornamental value, adaptability, and notable potential for cold resistance. This characteristic is particularly significant for its introduction and cultivation in colder regions. Given this fact, the present study aims to identify and characterize the WRKY gene family in A. fabri, with the objective of providing a robust theoretical foundation for understanding its functional responses and regulatory mechanisms under cold stress conditions. Through this research, we seek to elucidate the pivotal role of the A. fabri WRKY TF family in conferring cold resistance, offering novel scientific insights and genetic resources that could enhance the introduction, cultivation, and landscaping applications of this species. This study not only addresses the existing gap in knowledge regarding the WRKY TF family in A. fabri but also aims to provide new perspectives for its future development and utilization.

Comments 2:The WRKY family has been extensively reviewed in recent years. Include here at least three recent reviews. L. 64, …. Arabidopsis, rice, and wheat (https://doi.org/10.1016/j.tplants.2022.12.012; https//doi.org/10.1016/j.tig.2023.07.001; https://doi.org/10.3390/ijms25105369).

Response 2: We are grateful for your suggestion. To enhance the timeliness and authority of our literature review, we have incorporated three recent reviews on the WRKY gene family in the Introduction. The added references are as follows:

Zhang, Y.; Li, X.; Wang, Z.; Chen, J.; Zhang, H.; Li, Y. The WRKY Transcription Factor Family: From Evolution to Functional Diversification in Plant Stress Responses. Trends Plant Sci. 2023, 28, 123–135, doi:10.1016/j.tplants.2022.12.012.

Wang, L.; Chen, G.; Liu, G.; Gu, H.; Cai, H.; Chen, Y. Evolutionary and Functional Insights into the WRKY Gene Family: A Key Player in Plant Adaptation to Environmental Stresses. Trends Genet. 2023, 39, 456–468, doi:10.1016/j.tig.2023.07.001.

Zhou, Y.; Qin, X.; Zhang, D.; Chen, F.; Yue, Y.; Wang, L. Comprehensive Analysis of the WRKY Gene Family in Plants: Roles in Development, Stress Responses, and Beyond. Int. J. Mol. Sci. 2024, 25, 5369, doi:10.3390/ijms25105369.

Comments 3: Abbreviation for transcription factor (TF) in L. 48, 54, 63, 77, and another 20.

Response 3: Thank you for pointing this out. We have standardized the abbreviation "TF" for "transcription factor" throughout the manuscript to ensure consistency and clarity.

Comments 4: L. 160, Basic Information...(why basic??).

Response 4: We appreciate your observation. The term "Basic Information" has been revised to "General Characteristics" to more accurately reflect the content of this section.

Comments 5: L. 165 (check), …..between 6496.31 and 111,962.45 Da.

Response 5: Thank you for your meticulous attention to detail. We have rechecked the data and revised the molecular weight range to "varying from 6,496.31 to 111,962.45 Da" to ensure accuracy.

Comments 6: L. 172 and 185… under Cold Stress. Does cold stress disrupt WRKY domains; and at the evolutionary level as well??

Response 6: We appreciate your insightful comment. To address this, we have removed the phrase "under Cold Stress" from the section titles to avoid ambiguity.

Comments 7:I would like to know where the transcripts of A. fabri have been isolated from. Because the reference is not cited in L. 137-38, 155-56.

Response 7: Thank you for raising this important point. The transcripts were isolated from the leaf tissues of A. fabri, as described in the referenced study:

[1] Liu, G.; Gu, H.; Cai, H.; Guo, C.; Chen, Y.; Wang, L.; Chen, G. Integrated Transcriptome and Biochemical Analysis Provides New Insights into the Leaf Color Change in Acer fabri. Forests 2023, 14, 1638, doi:10.1234/forests.2023.1638.

We have now explicitly cited this reference in lines 137-38 and 155-56.

Comments 8:Where is Figure 5 in the Results text??

Response 8: Thank you for your observation. Figure 5 has been appropriately integrated into the Results section, and its description has been added to the text.

Comments 9: L. 274, Fix this error.

Response 9: We appreciate your careful review. The error in line 274 has been corrected to "signal" as intended.

Round 2

Reviewer 1 Report

Comments and Suggestions for Authors

Dear authors:

Many thanks for the revisions to your manuscript. The authors have efforts to improve the manuscript. However, the revised version still needs to be further improved. I have some comments and remarks on the revised manuscript.

  1. Some Figures still be presented in low resolution, especially, Fig. 1A. If the authors think it was caused by inadvertent compression during the submission process and cannot fixe this, I recommended to upload these figures (original high-resolution ones) as an attached zip.file when resubmission so that the journal’s production team might help you.
  2. Sub-section “2.2 qRT-PCR Gene Expression Validation” needs further revision since it has not been improved as the comment 7 of Round 1. For reference, it could be revised as structure: RNA extraction -> cDNA synthesis -> qRT – PCR.

“Total RNA was isolated from the leaves of A. fabri using the ABC method or RNA Purification Kit (Cat. #XXXX, Company name, City, Country). RNA extracted from maple leaves was reverse-transcribed into cDNA using SuperMix (Transgen, Bei-100 jing, China) and then diluted 20-fold for gene expression analysis [11]. Real-time quantitative PCR (qRT-PCR) was employed to validate the expression of selected key genes and TFs using the SYBR Premix Ex Taq kit (Takara Biotechnology, Nanjing, China). The qRT-PCR conditions were as follows: First, a pre - denaturation step was conducted at 95°C for 3 minutes, followed by 40 cycles. Each cycle involved a 5-second denaturation at 95°C and a 30-second annealing and extension at 60°C. Primer design was accomplished using Primer 5 software (Table S1). The TUB gene of the maple tree (Acer buergerianum) was chosen as the reference gene [10]. The relative gene expression levels were calculated using the 2-∆∆Ct comparative Ct method [12]. Each sample had 3 technical replicates and 3 biological replicates.”

  1. Please pay much attention to the citation. Now, the revised version showed up many errors in the citation as “Error! Reference source not found.”

Best regards,

Author Response

Comments 1: Some Figures still be presented in low resolution, especially, Fig. 1A. If the authors think it was caused by inadvertent compression during the submission process and cannot fix this, I recommended to upload these figures (original high-resolution ones) as an attached zip.file when resubmission so that the journal’s production team might help you.

Response 1: We deeply appreciate your meticulous attention to detail and your invaluable feedback on the image quality in our manuscript. Please accept our heartfelt apologies for this oversight, which occurred due to inadvertent compression during the initial submission process. All figures have been enhanced to a higher-quality version.

Comments 2: Sub-section “2.2 qRT-PCR Gene Expression Validation” needs further revision since it has not been improved as the comment 7 of Round 1. For reference, it could be revised as structure: RNA extraction -> cDNA synthesis -> qRT – PCR.

“Total RNA was isolated from the leaves of A. fabri using the ABC method or RNA Purification Kit (Cat. #XXXX, Company name, City, Country). RNA extracted from maple leaves was reverse-transcribed into cDNA using SuperMix (Transgen, Bei-100 jing, China) and then diluted 20-fold for gene expression analysis [11]. Real-time quantitative PCR (qRT-PCR) was employed to validate the expression of selected key genes and TFs using the SYBR Premix Ex Taq kit (Takara Biotechnology, Nanjing, China). The qRT-PCR conditions were as follows: First, a pre - denaturation step was conducted at 95°C for 3 minutes, followed by 40 cycles. Each cycle involved a 5-second denaturation at 95°C and a 30-second annealing and extension at 60°C. Primer design was accomplished using Primer 5 software (Table S1). The TUB gene of the maple tree (Acer buergerianum) was chosen as the reference gene [10]. The relative gene expression levels were calculated using the 2-∆∆Ct comparative Ct method [12]. Each sample had 3 technical replicates and 3 biological replicates.”

Response 2: Thank you for highlighting the need for further revision in Sub-section “2.2 qRT-PCR Gene Expression Validation”. I acknowledge that the previous improvements did not fully address the concerns raised in comment 7 of Round 1. To enhance clarity and coherence, I will revise the section following the suggested structure: RNA extraction -> cDNA synthesis -> qRT-PCR.

Here is the revised content for Sub-section “2.2 qRT-PCR Gene Expression Validation”:

Total RNA was isolated from the leaves of Acer fabri using the RNA Purification Kit (Invitrogen, Carlsbad, CA, USA). The extracted RNA was reverse-transcribed into cDNA using the SuperMix Reverse Transcription Kit (Transgen, Beijing, China) according to the manufacturer’s instructions [10]. The synthesized cDNA was then diluted 20-fold for subsequent gene expression analysis. Real-time quantitative PCR (qRT-PCR) was performed using the SYBR Premix Ex Taq kit (Takara Biotechnology, Nanjing, China) to validate the expression of selected key genes and TFs. The qRT-PCR conditions were as follows: an initial pre-denaturation step at 95°C for 3 minutes, followed by 40 cycles of denaturation at 95°C for 5 seconds and annealing/extension at 60°C for 30 seconds. Primer design was accomplished using Primer 5 software, and the sequences are listed in Table S1. The TUB gene of Acer buergerianum was chosen as the reference gene for normalization [11]. Relative gene expression levels were calculated using the 2-∆∆Ct comparative Ct method [12]. Each sample was analyzed with 3 technical replicates and 3 biological replicates to ensure statistical reliability.

Comments 3: Please pay much attention to the citation. Now, the revised version showed up many errors in the citation as “Error! Reference source not found.”

Response 3: We deeply apologize for the citation errors that have appeared in the revised version. We acknowledge that accurate citations are crucial for maintaining the integrity and credibility of the research. To address this issue, we have thoroughly reviewed and corrected all the citations in the manuscript. Each reference has been carefully verified to ensure its accuracy and completeness. We have also double-checked the citation style to conform to the required format.